# Socio-Occupational and Health Conditions in Intensive Care Unit (ICU) Professionals during the COVID-19 Pandemic in Spain

**Fernanda Gil-Almagro** [1,2,*], **Fernando Jose García-Hedrera** [1], **Francisco Javier Carmona-Monge** [3], **Cecilia Peñacoba-Puente** [2] **and Patricia Catalá-Mesón** [2]

1    Critical Care Unit, Hospital Universitario Fundación Alcorcón, 28922 Alcorcón, Madrid, Spain
2    Department of Psychology, Universidad Rey Juan Carlos, 28922 Alcorcón, Madrid, Spain
3    Servicio de Anestesiología y Reanimación, Complejo Hospitalario Universitario de Santiago de Compostela, 15706 Santiago de Compostela, A Coruña, Spain
*    Correspondence: fgilalmagro@gmail.com

**Abstract:** Objective: The aim of this research is to analyze the socio-occupational and health conditions of Intensive Care Units (ICU) health professionals during the COVID-19 pandemic in Spain. In addition, with regard to the working conditions (availability of personal protective equipment—PPE, workload and patient/professional ratio), this research aims to analyze the possible differences depending on the Spanish region that was sampled and their professional category, as well as their relationship with the characteristic symptoms of COVID-19 (myalgias and respiratory distress). Method: A cross-sectional study performed with an online questionnaire, which was spread throughout all of the Spanish autonomous communities/regions. Results: The sample consisted of 461 ICU professionals in Spain, of whom, 94% reported an increase in their workload, and 43% reported a patient/professional ratio that was higher than it usually is. The median professional experience in the ICU was 9.73 years, with 47% of them having less than 5 years of experience in it. About 80% had undergone some diagnostic tests. There is a significant difference in 'Serology (+)' in terms of sex, with males having a serology (+) in 26% of the cases and females having it in 13% of the cases ($p = 0.011$). Most of the professionals (80%) were concerned about a possible infection, and up to 96% were worried about infecting their family members. The most common COVID-19 symptom was a headache, with there being a higher incidence of this in women. Significant differences were observed with respect to the availability of appropriate PPE depending on the Spanish region (i.e., Cataluña had best rate of PPE availability) ($p = 0.005$). The higher incidence of myalgias and respiratory distress were associated with a lower availability of PPE and a higher patient/professional ratio. Conclusions: The ICU staff reported an increase in their workload, with an increase in the amount of staff who had less experience. A high percentage of them have suffered symptoms, although the proportion of positive tests was low. The most characteristic symptoms of COVID-19 seem to be related to the working conditions. The results show the socio-occupational and health conditions of Spanish ICU professionals during the pandemic and point to the need to establish occupational risk-prevention measures.

**Keywords:** COVID-19; intensive care units; health professionals; Spain; socio-occupational variables; work overload

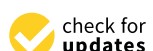



## 1. Introduction

On 31 December 2019, the Wuhan Municipal Health and Sanitation Commission (Hubei province, China) reported a cluster of 27 cases of pneumonia of unknown etiology with the initial onset of the symptoms occurring on 8 December, including seven severe cases, without them being able to identify the source of the outbreak. On the 7 January 2020, the Chinese authorities identified a new type of RNA virus of the Coronaviridae family as

the causative agent of the outbreak, which they named Severe Acute Respiratory Syndrome coronavirus 2 (SARS-CoV-2). It is transmitted by airborne transmission mechanisms (respiratory droplets) and through contact of the virus with the conjunctiva, nasal and oral mucosa [1,2].

The International Health Regulations Emergencies Committee declared the outbreak a public health emergency of international concern on the 30 January 2020. Subsequently, on 11 March 2020, the World Health Organization (WHO) recognized it as a global pandemic [3].

In Spain, as of the 28 February, 32 cases had been reported [4]. Subsequently, due to an increase in the number of cases and a saturation of the health system that could lead to its collapse, a state of alarm was declared on the 13 March, which led to the population being confined to their homes for more than two months, ending on 21 June. Throughout this time, the surveillance of the cases of this disease in Spain had been based on the universal notification of confirmed cases of COVID-19 which were identified in each region (autonomous community). The number of cases increased exponentially over the months, reaching 1,291,808 cases on 18 November 2020, with the principal outbreaks of the cases being in the Community of Madrid (CM) (268,405) and Cataluña (234,421) [5]. An estimated 5.6% of the identified cases were hospitalized, 0.4% of them required ICU treatment and 1.0% of them died. The highest proportion of the COVID-19 cases occurred in the 15–59 age group (68% of the total), with the 15–29 age group being the most represented, accounting for 21% of the total. In addition, the percentage of hospitalizations and deaths that were due to COVID-19 increased with age, reaching 26.7% (hospitalizations) and 10.4% (deaths) in those that were over 79 years of age [6].

All of this has resulted, over the course of these months, in a situation whereby, in all of the affected countries, they are close to the collapse of their healthcare systems (worldwide, as of June 9th, there have been around 7,039,918 infections, with a total of 404,396 deaths), thereby causing a significant overload for healthcare personnel [7]. Although the entire healthcare system has been overwhelmed in general, the Intensive Care Units (ICUs) have been particularly affected by the COVID-19 pandemic, placing a significant burden not only physically but also psychologically on the healthcare staff that are working in these units.

Health professionals that are directly involved in the diagnosis, treatment and care of these patients during this pandemic have been subjected to a significant increase in their workload, with there being numerous cases of COVID-19-infected professionals among those who have treated these patients. The physical and psychological integrity of one's health is at high risk when one is working under such stressful conditions. In addition, factors such as the reduction in the ratio of staff with sufficient professional experience in the ICU, the shortage of personal protective equipment (PPE), the absence of an effective treatment for the disease and the presence of feelings of not having adequate support have directly contributed to the psychological burden on the staff who have treated these patients [8]. Health professionals have been one of the fundamental pillars in containing the pandemic, despite it having a high cost to their health on many occasions [9]. In Spain, it is known that the number of health professionals that have been infected has been one of the highest in the world, with some 41,000 health professionals being infected, accounting for approximately 14% of the total number of infected people [10]. In any case, various factors may have played a role in this (a high burden of care, a shortage of personal protective equipment in the early stages of the crisis, etc.), although we cannot be sure what was the main reason for this high number of infections. According to the data that were provided by the National Epidemiological Surveillance Network, 10% of the healthcare professionals who were diagnosed required hospital admission, and there were more women affected by COVID-19 than there were men. However, men had a higher incidence of having pneumonia and underlying diseases, and there was a higher percentages of hospitalizations, admissions to the ICU and mechanical ventilation among men than there were among women. Of these, 310 (1.1%) required admission to the ICU and, fortunately, the mortality rate was 0.1% (53 cases as of 21 May) [10].

There are several studies in the literature that analyze both the physical and psychological effects of severe infectious disease outbreaks on healthcare professionals, which are related to work overloads and stress that are caused by the high risk of infection. Studies conducted in other situations with similar characteristics (the SARS outbreak in 2003, the H1N1 outbreak in 2009 and the MERS outbreak in 2012) have demonstrated the existence of dysfunctional psychological reactions among these professionals [11–13]. Among the most relevant findings revealed the health professionals' fear of infecting their family or friends, the fear of uncertainty and stigmatization, as well as high levels of stress, anxiety, and depressive symptoms, which could have long-term psychological effects [11,12]. Additionally, a two-to-three-times higher risk of developing post-traumatic stress disorder was observed among professionals who were in contact with SARS patients than among those who did not have such exposure [11–13]. Several studies carried out in the Wuhan region on healthcare workers who have cared for COVID-19 patients have detected an increase in the psychological overload of the workers who carry out the direct care tasks, especially among female nurses [14]. In addition, a high level of concern on the part of the professionals has also been reported in relation to the possibility of infection to other members of the family unit, thereby generating high levels of anxiety and stress as a consequence [15,16].

At present, most of the data regarding the health of the professionals who have cared for patients with COVID-19 have been carried out on Chinese populations, with there being a little amount of data that are currently available in our setting. Furthermore, most of the studies have focused on specifying the epidemiology and clinical characteristics of the infected patients, as well as the genomic characterization of the virus, or the economic and social challenges that this pandemic presents for the governments of the affected countries. However, even though the health sector is the most affected by this pandemic, there are still not any known studies in Spain describing the socio-occupational and health conditions of our professionals during these months.

The main objective of this study was to analyze the socio-occupational and health conditions of health professionals who have worked in the ICU during the first wave of the pandemic. Specifically, the sociodemographic variables such as age, sex, marital status and aspects which are related to the place of residence or the Spanish autonomous community, among others, were analyzed. Regarding the occupational variables, the years of experience in the ICU, the professional category, the availability of PPE and the workload or the patient/professional ratio are of special interest. In relation to the health issues, the PCR and serology were analyzed as well as the symptoms that are associated with COVID-19.

In addition, with regard to the working conditions (availability of PPE, workload and patient/professional ratio), this research aimed to analyze the possible differences depending on the Spanish region that was sampled and the professional category, as well as their relationship with the characteristic symptoms of COVID-19 (myalgias and respiratory distress).

## 2. Methodology

### 2.1. Design

This is a cross-sectional descriptive study at a national level (Spain), including the total of the 17 autonomous communities (Andalucía, Aragón, Principado de Asturias, Illes Balears, Canarias, Cantabria, Castilla y León, Castilla-La Mancha, Cataluña, Comunitat Valenciana, Extremadura, Galicia, CM and Murcia) and two autonomous cities (Ceuta and Melilla). The study subjects were the healthcare professionals (doctors, nurses and nursing care technicians), who have provided their services in the ICUs where the COVID-19 patients have been admitted during the months of March to June 2020. The STROBE guideline for cross-sectional studies has been followed (see Table S1).

## 2.2. Study variables

We reviewed the previous surveys that were carried out both for this pandemic and for previous outbreaks of other infectious diseases (SARS, MERS and H1N1) [11–13,17]. Information (using an ad hoc questionnaire) was collected on: socio-demographic and occupational data (occupational category, sex, age, educational level, the place of residence during the pandemic, marital status, the number of hours that were worked, the work shift type and whether they lived in a household with children), presence of physical symptoms that are related to a COVID-19 infection during the study period, data on the need for health care during the study period (attendance at a medical consultation, hospitalization, isolation or diagnostic tests having occurred), concern about the possibility of infection of oneself or other family members with COVID-19, and data regarding the working conditions (experience of the COVID-19 infection, experience of hospitalization, isolation or diagnostic tests having occurred), and data regarding the possibility of infection of oneself or other family members with COVID-19 (hospitalization, isolation or diagnostic tests having occurred), concern about the possibility of infection of oneself or other family members with COVID-19, and data on the working conditions (the work experience in the unit, the patient-worker ratio, the workload during the pandemic and the availability of appropriate PPE).

## 2.3. Sample Selection

The sample was selected by means of non-probabilistic snowball and convenience sampling method, firstly by contacting professionals that were dedicated to the care of critical patients who were known to the research team and requesting their dissemination in their units or hospital centers. In addition, the dissemination was also carried out through social networks (Facebook, Twitter and LinkedIn), selecting those professionals whose profile stated that they were working in intensive care and resuscitation units; if they responded, they were asked to send the link to other professionals who met our inclusion criteria. In addition, the link to the survey was sent with a cover letter to the communication offices or research committees of the hospital centers with a request to disseminate the address of the survey via the corporate mail. The study included healthcare professionals from the Spanish healthcare system who provided their services in resuscitation and critical care units in which COVID-19 patients were admitted during the months from March to June 2020, and it excluded those professionals who were not doctors, nurses or nurses' assistants, and those professionals who did not undertake healthcare tasks. The required sample size was calculated based on the expert recommendations that suggest a minimum of 300 to 450 participants to obtain sufficiently comparable patterns in the data analysis [18].

## 2.4. Data Collection

The data collection was carried out using an online electronic form that was designed for this purpose by the research team. On the first page of the questionnaire, the participants were informed of the aim of the research and were asked to give their consent for the data to be used in the study. Once they agreed to take part in the study, they submitted a form on which the information relating to the variables that are described above was collected.

## 2.5. Ethical Considerations

The study was approved by the hospital's Clinical Research and Ethics Committee and the informed consent was obtained from the participants before the questionnaire was administered. In addition, the present work has received the scientific endorsement of the Spanish Society of Intensive Care Nursing and Coronary Units (SEEIUC). This study was conducted in accordance with the national and international guidelines of the code of ethics, the declaration of Helsinki and the code of good practice and SAS Order 3470/2009. The processing of the patients' personal data which were collected in this study complied with Organic Law 15/1999 of 13 December on the Protection of Personal Data (LOPD) and with Regulation (EU) No. 2016/679 of the European Parliament and of the Council of

27 April 2016 on Data Protection (GDPR). All of the information that were collected, stored and processed are anonymous.

### 2.6. Statistical Analysis

The qualitative variables were described by frequencies and percentages (%). The normality of the quantitative variables was determined with the Kolmogorov–Smirnov test, and the analysis of the symmetry of the distribution and the proximity between the mean and the median were conducted. The quantitative variables were described as the mean (standard deviation) for the normally distributed variables and as the median (interquartile range) for the variables with a non-normal distribution. In the case of the categorical variables, the $\chi^2$ test was used to analyze their association with the independent variables. A statistical analysis was carried out using the Statistical Package for the Social Sciences (SPSS) version 21 for Windows (IBM, Armonk, NY, USA). The results were considered statistically significant when $p < 0.05$.

## 3. Results

### 3.1. Participants

This study is part of a larger project that aimed to analyze the health and working conditions of the health personnel in Spain during the COVID-19 pandemic (specifically between 16 May 2020 and 30 June 2020). A total of 1466 health care workers participated in this project, who previously signing the corresponding informed consent form. Of these, for this particular study, those who had performed their services in the ICU were selected.

Finally, the study sample was comprised of 461 ICU professionals (31.73%) (Figure 1).

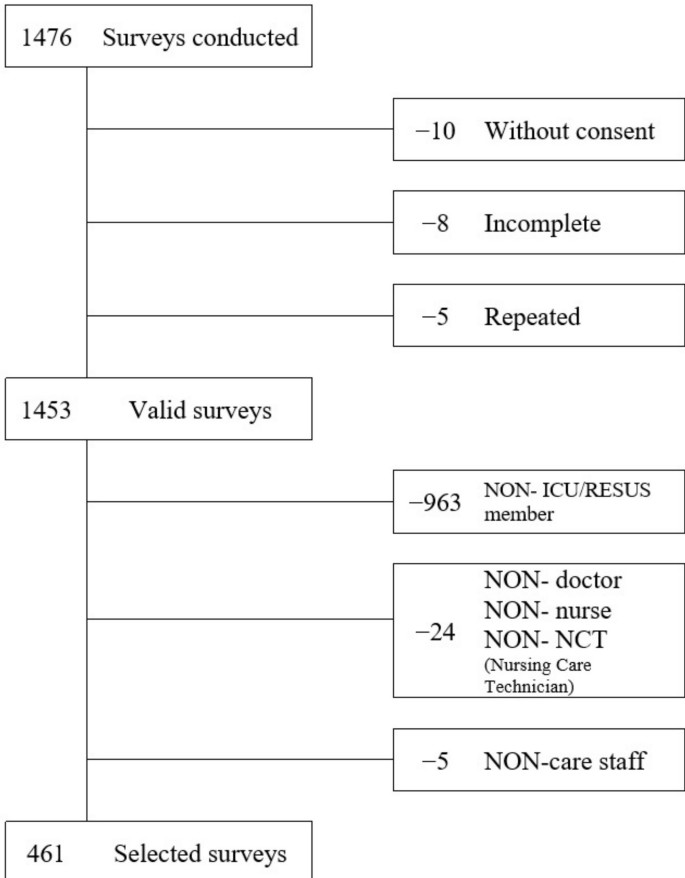

**Figure 1.** Study sample selection.

### 3.2. Socio-Occupational and Health Conditions of ICU Professionals

Table 1 shows the data on the socio-demographic characteristics of the participants. In our sample, the age ranged between 18 to 64 years, and the mean age was 39.68 years (SD = 10.57). Most of the participants were women (85%), and more than half of them did not have children or did not live with them during the pandemic. Most of the sample had completed university studies (83%). Almost a third of the sample worked in the ICUs in CM (53%) or Cataluña (20%). The vast majority of the professionals resided in their usual residence during the pandemic situation (91%), and more than half of the sample (61%) lived with a partner.

**Table 1.** Socio-demographic characteristics of the sample.

|  |  | *n* (%) |
|---|---|---|
| Age (years) | ≤30 | 110 (23.8%) |
|  | 30–40 | 127 (27.5%) |
|  | 41–50 | 138 (29.9%) |
|  | >50 | 86 (18.7%) |
| Sex | Female | 391 (84.8%) |
| Marital status | Single | 151 (32.7%) |
|  | In a relationship | 280 (60.7%) |
|  | Separated | 28 (6.1%) |
|  | Widowed | 2 (0.4%) |
| Academic level | Basic | 80 (17.3%) |
|  | University | 257 (55.7%) |
|  | Postgraduate | 124 (26.9%) |
| Residence during the pandemic | Usual residence | 421 (91.3%) |
|  | Hotel | 11 (2.4%) |
|  | Other | 29 (6.3%) |
| Household members | 1 | 107 (23.2%) |
|  | 2 | 121 (26.2%) |
|  | 3–5 | 230 (49.9%) |
|  | ≥6 | 3 (0.6%) |
| Living with children during the pandemic | No children/no children living in the household | 260 (56.4%) |
|  | Children <16 years old in the household | 130 (28.2%) |
|  | Children >16 years old in the household | 50 (10.8%) |
|  | Children < and >16 years old in the household | 21 (4.6%) |
| Autonomous community | Comunidad de Madrid | 246 (53.3%) |
|  | Cataluña | 87 (18.9%) |
|  | Rest of Spain | 128 (27.8%) |

The mean value in relation to the years of experience that they had in the unit was 9.73 (SD = 9.17), which ranged between less than one year to 35 years (data not shown). Half of the participants (53%) had six years or more of experience in the unit. The number of professionals with less than one year of experience was much higher in the groups of nurses and nursing care technicians than it was in the case of the doctors (data not shown). Most of the sample (69%) were nurses, and half of the professionals had a permanent contract (Table 2). Of the professionals who maintained a provisional or temporary contractual relationship with the institution, 33% reported having been transferred to the ICU from other units during the pandemic and 14% of them reported having been hired during this period (data not shown). Practically all of the professionals reported having worked in units that have been expanded beyond their capacity, and 43% indicated that there was a higher-than-usual patient/professional ratio. Almost half of the professionals (46%) indicated the unavailability of an adequate availability of PPE.

**Table 2.** Employment characteristics of the sample.

|  |  | n (%) |
|---|---|---|
| Unit experience (in years) | <1 | 69 (15%) |
|  | 1–5 | 148 (32.1%) |
|  | ≥6 | 244 (52.9%) |
| Professional category | Doctor | 61 (13.2%) |
|  | Nurse | 319 (69.2%) |
|  | NCT | 81 (17.6%) |
| Employment relationship | Permanent | 227 (49.2%) |
|  | Intern | 98 (21.3%) |
|  | Temporary | 136 (29.5%) |
| Work shifts | Fixed shift | 62 (13.5%) |
|  | Rotating shifts | 178 (38.7%) |
|  | Shifts + on call duty | 46 (10%) |
|  | 12/24 h | 174 (37.8%) |
| Higher-than-usual patient/professional ratio | Yes | 198 (42.9%) |
| Higher-than-usual workload | Yes | 434 (94.1%) |
| Availability of adequate PPE | Yes | 247 (53.6%) |

NCT: nursing care technicians; PPE: personal protective equipment.

Regarding their health status, prior to the COVID-19 pandemic, 20% of the participants indicated that they had been diagnosed with a chronic disease (data not shown). During the pandemic, 18% of them reported having been to a doctor's office, and 16% of them had to undergo some form of isolation, and only two professionals required hospitalization (data not shown). Approximately 20% of the participants reported having had contact outside of work with people who were infection with COVID-19 (data not shown).

Most of the sample (80%) were concerned about a possible infection occurring (data not shown), with this percentage increasing (96%) in relation to the concern about them infecting their family members (data not shown).

Table 3 shows the PCR and serology test results of the professionals who have undergone any diagnostic test. There is a significant difference ($p = 0.014$) in the 'Serology (+)' data in relation to sex, with the males having had a serology (+) in 26% of the cases, while the females had a serology (+) in 12% of the cases.

**Table 3.** PCR and serology tests performed on the professionals participating in the study.

|  |  | Male | Female |  |  |
|---|---|---|---|---|---|
|  | n (%) | n (%) | n (%) | $\chi^2$ | *p* |
| COVID testing during the pandemic | 368 (79.8%) | 57 (81.4%) | 311 (79.5%) | 0.132 | 0.872 |
| PCR (+) | 38 (10.3%) | 8 (14%) | 30 (9.6%) | 1.002 | 0.343 |
| PCR (−) | 261 (70.9%) | 44 (77.2%) | 217 (69.8%) | 1.285 | 0.341 |
| Serology (+) | 55 (14.9%) | 15 (26.3%) | 40 (12.9%) | 6.859 | **0.014** |
| Serology (−) | 251 (68.2%) | 32 (56.1%) | 219 (70.4%) | 4.528 | **0.044** |

Regarding the symptoms that are associated with COVID-19, only 12% of them reported a total absence of symptomatology (data not shown). Table 4 shows the data on the symptomatology which were reported by the professionals, and the distribution of this by sex. Statistically significant differences can be observed between the men and women in relation to the occurrence of headache, dizziness, skin manifestations, a fever, ageusia and anosmia, with there being higher percentages among the women in the first three symptoms and there being higher percentages for the men in the last three.

**Table 4.** COVID-19-related symptomatology present in the sample.

| | | Male | Female | | |
|---|---|---|---|---|---|
| | **n (%)** | **n (%)** | **n (%)** | **$\chi^2$** | ***p*** |
| Headache | 345 (74.8%) | 42 (60%) | 303 (77.5%) | 9.648 | **0.002** |
| Sore throat | 193 (41.9%) | 23 (32.9%) | 170 (43.5%) | | |
| Myalgias | 167 (36.2%) | 23 (32.9%) | 144 (36.8%) | | |
| Cough | 152 (33%) | 24 (34.3%) | 128 (32.7%) | | |
| Rhinitis | 99 (21.5%) | 15 (21.4%) | 84 (21.5%) | | |
| Dizziness | 93 (20.2%) | 8 (11.4%) | 85 (21.7%) | 3.919 | **0.048** |
| Chills | 88 (19.1%) | 11 (15.7%) | 77 (19.7%) | | |
| Chest pain | 88 (19.1%) | 8 (11.4%) | 80 (20.5%) | | |
| Fever | 66 (14.3%) | 18 (25.7%) | 48 (12.3%) | 8.740 | **0.003** |
| Skin manifestations | 65 (14.1%) | 2 (2.9%) | 63 (16.1%) | 8.613 | **0.003** |
| Respiratory distress | 56 (12.1%) | 8 (11.4%) | 48 (12.3%) | | |
| Ageusia | 37 (8%) | 10 (14.3%) | 27 (6.9%) | 4.381 | **0.036** |
| Anosmia | 35 (7.6%) | 11 (15.7%) | 24 (6.1%) | 7.760 | **0.005** |

*3.3. Relationship between Working Conditions (Availability of PPE, Workload and Patient/Professional Ratio) and Spanish Region, Professional Category and COVID-19 Symptoms (Myalgias and Respiratory Distress)*

Table 5 shows the results of the relationship between the occupational risk factors that were considered (the availability of PPE, the work overload and a higher-than-usual patient/ professional ratio) and certain sociodemographic variables (region and professional category) and the most characteristic symptoms of COVID-19 (myalgias and respiratory distress). As far as the sociodemographic variables are concerned, the results show that there were better conditions in Cataluña compared to the rest of the regions that were analysed, especially in terms of the greater availability of PPE ($p = 0.002$) and the patient/professional ratio ($p = 0.001$). No statistically significant differences were observed between the occupational risk factors that were considered and the professional categories. Regarding the most characteristic symptoms of COVID-19 (myalgias and respiratory distress), statistically significant relationships were observed, both for the availability of PPE ($p < 0.001$ and $p = 0.027$, respectively) and for the patient/professional ratio ($p = 0.022$ and $p = 0.010$, respectively). The percentage of both of the symptoms was higher when the PPE was not available and when the patient/professional ratio was higher than that which is usual.

**Table 5.** Analysis of differences in clinical and occupational variables as a function of perceived PPE availability, patient/professional ratio and workload during the pandemic.

| | | Availability of PPE | | | | Ratio Higher than Usual | | | | Workload Higher than Usual | | | |
| --- | --- | --- | --- | --- | --- | --- | --- | --- | --- | --- | --- | --- | --- |
| | | NO | YES | $\chi^2$ | *p* | NO | YES | $\chi^2$ | *p* | NO | YES | $\chi^2$ | *p* |
| | | n (%) | n (%) | | | n (%) | n (%) | | | n (%) | n (%) | | |
| Region | Comunidad de Madrid | 128 (52%) | 118 (48%) | 12.69 | **0.002** | 122 (49.6%) | 124 (50.4%) | 14.752 | **0.001** | 9 (3.7%) | 237 (96.3%) | 11.52 | **0.021** |
| | Cataluña | 26 (29.9%) | 61 (70%) | | | 63 (72.4%) | 24 (27.6%) | | | 4 (4.6%) | 83 (95.4%) | | |
| | Rest of Spain | 60 (46.9%) | 68 (53.1%) | | | 78 (60.9%) | 50 (39.1%) | | | 14 (10.9%) | 114 (89.1%) | | |
| Professional category | Doctor | 28 (45.9%) | 33 (54.1%) | 2.505 | 0.286 | 34 (55.7%) | 27 (44.3%) | 5.599 | 0.061 | 7 (11.5%) | 54 (88.5%) | 5.961 | 0.202 |
| | Nurse | 142 (44.5%) | 177 (55.5%) | | | 192 (60.2%) | 127 (39.8%) | | | 18 (5.6%) | 301 (94.4%) | | |
| | NCT | 44 (54.3%) | 37 (45.7%) | | | 37 (45.7%) | 44 (54.3%) | | | 2 (2.5%) | 79 (97.5%) | | |
| Myalgias | No | 118 (40.1%) | 176 (59.9%) | 12.888 | **<0.001** | 179 (60.9%) | 115 (39.1%) | 4.87 | **0.027** | 18 (6.1%) | 276 (93.9%) | 0.104 | 0.747 |
| | Yes | 96 (57.5%) | 71 (42.5%) | | | 84 (50.3%) | 83 (49.7%) | | | 9 (5.4%) | 158 (94.6%) | | |
| Respiratory distress | No | 180 (44.4%) | 225 (55.6%) | 5.236 | **0.022** | 240 (59.3%) | 165 (40.7%) | 6.642 | **0.010** | 25 (6.2%) | 380 (93.8%) | 0.604 | 0.437 |
| | Yes | 34 (60.7%) | 22 (39.3%) | | | 23 (41.1%) | 33 (58.9%) | | | 2 (3.6%) | 54 (96.4%) | | |

NCT: nursing care technicians; PPE: personal protective equipment.

## 4. Discussion

The COVID-19 pandemic has tested our healthcare system. There has been an overload on the care that is provided at all levels (particularly important in the ICU) which were previously unknown to the professionals. However, the impact of this and the perception of the overload has not been the same in different territories or among the different professional groups providing the services in these units. This paper analyzes this impact on the ICU professionals in Spain, contemplating its different regions.

A very important fact that was obtained from our work is that a large percentage of professionals who have worked in the ICU during the pandemic, specifically a quarter of the participants, are under the age of 30, who had scarce work experience in the ICU. Likewise, we also found a high percentage of professionals who reported having been transferred to the ICU from other units during the pandemic due to the expansion of these units, as well as the decrease in activity in the rest of the hospital. Based on the results that were obtained, it can be said that the proportion of expert staff, which is defined by Benner as those who have an intuitive command of clinical practice that gives them the ability to identify a problem with little or no time being wasted on alternative solutions, was very low [19]. Additionally, it is necessary to take into account the percentage, according to our results of around 15%, of the new professionals who suppose, in turn, an added overload for the expert professionals [20,21]. In this line, Robinson et al. points out the importance of the expert nurse in an intensive care setting as they will be able to solve the problems of high complexity, achieve a continuous improvement in the activity of the work team and provide a high quality of care resulting in the satisfaction of both the patients and their families [22]. The SEEIUC itself has already pointed out the importance of transferring resources from the less affected areas to those with an evident overload in the ICUs, as the learning process of a critical care nurse requires months of postgraduate training [23].

Another issue of interest in this study is the overload of the healthcare workers in the ICU during this initial period of the pandemic. Our data indicate that the unfavorable working conditions (the work overload, the patient/professional ratio and non-availability of PPE) have been important risk factors, with no significant statistical differences being observed depending on the professional category that was being performed. It is important, in turn, to differentiate between the perception of an overload and the more objective indicators that are related to the patient/professional ratio. Thus, our data indicate that barely half of the participants reported a higher-than-usual patient/professional ratio, however, practically all of them (94%) reported a higher-than-usual workload. In any case, we must not overlook the fact that work overload may be due not only to the high number of patients and their serious clinical situation, and the associated uncertainty, but may also be influenced by a number of unrelated circumstances, such as the use of PPE for a large number of hours by professionals who were providing direct care to these patients, and it may also be aggravated by the number of professionals with little work experience in the care of critical patients [24]. In this sense, it is also interesting to highlight, given the Spanish health system (each autonomous community/region is responsible for providing health services through its own health service), the inequality that is observed in relation to the availabilities of PPE, the patient/professional ratios and the workloads between the different regions. Our data indicate that CM has had the most unfavorable conditions when it is compared to the rest of the Spanish regions.

Thus, as previously pointed out, during the pandemic, a high proportion of the staff in the intensive care units were made up of professionals with less than one year's experience, and given the seriousness of the situation, it was not possible to find "expert" staff in these areas to cover the expansion of the ICUs. This may have contributed to an increased perception of the workload. The importance of having a qualified team for the care of the critical patients has been highlighted by the different scientific societies, as well as by the Ministry of Health itself, with the aim not only of offering an improvement in the quality of patient care, but also of providing greater safety in the patient care that is given. In fact, the

current health crisis has once again highlighted the need for a critical care nursing specialty, with its own entity and that which is independent from its more general Medical–Surgical Care specialty [25]. The additional concern about the contagion may have contributed to this overload. Furthermore, in our sample, although most of the healthcare workers were concerned about a possible infection occurring, almost all of them were very worried about the possibility of infecting their family members, which may have generated additional stress among them. Similar results have been found in other countries with a high incidence of the disease in both the general population and the healthcare workers [26,27].

Another aspect of interest that this study analyzes is in relation to the symptoms that are associated with COVID-19 in the ICU professionals, given that in Spain there has been a high number of infections among the health professionals [10]. However, our results indicate that, despite this high incidence, only one of the ICU participants reported the occurrence of a hospitalization and a low percentage of them had gone for medical consultation. In our ICU sample, the proportion of infected professionals was around 10%, and about 15% of them had generated immunity to the coronavirus, finding statistically significant differences that were in favor of the males in relation to immunity. It is important to emphasize this situation, given that despite the high number of professionals with no previous experience in the care of critical patients and the perceived low availability of PPE in some of the centers, the fact that they knew from the outset that they were caring for coronavirus-positive patients led them to take extreme protective measures with the means that were available, thereby avoiding exposure to the pathogen.

Nevertheless, although the percentage of those who were infected was low, almost all of the professionals reported having suffered some type of symptom during the pandemic. The most frequently reported symptoms were the more non-specific ones which, although they appear when one is infected by this micro-organism, can be associated with many other situations that are unrelated to it. Specifically, the most frequently reported symptom was a headache, which was present in three quarters of the ICU professionals, and we observed that there were statistically significant differences that indicate that there was a higher incidence among the women. This fact, however, may be related to multiple other situations that occurred during this period, such as having an excessive workload, the high number of shifts or the stress that is derived from the care of such patients. In fact, Ong et al. found that up to 81% of the professionals reported having suffered headaches or aggravations of their usual headaches, which were probably associated with the use of PPE continuously for more than 4 h during the shift [28]. On the other hand, we must take into account that a part of this symptomatology may be derived from a period of home confinement, which in the case of health professionals added to the situation of the work-related stress. Different studies have shown the role of this situation of confinement in the onset of non-specific physical symptoms, as well as the appearance of anxiety and stress in the general population, and so we cannot rule out the possibility that something similar has happened in the ICU workers [29–31]. The results are similar to those that were obtained by Cao et al. [32] and Chew et al. [33] who found that practically all of the professionals presented mild body discomfort, including symptoms such as tiredness, sore throat, cough, neck, shoulder and back pain, nausea and even skin alterations, with none of the participants being infected by coronavirus.

As opposed, the most characteristic symptoms of the COVID-19 disease, such as respiratory distress or fever, appeared less frequently among the ICU professionals in our results. As they are relevant data, our results indicate that there were statistically significant differences in the most characteristic symptoms of COVID-19 (respiratory distress or fever) depending on the availability of PPE and the increase in the patient/professional ratio that was above the usual value.

Regarding the limitations of the present study, perhaps the most important is a possible selection bias that is related to the sampling methodology, which was used to select the sample, in such a way that the presence of some professional categories may have been overestimated. Likewise, the difficulty of contacting the different intensive care units

in Spain meant that some of the territories were overrepresented (mainly those which were most affected by the coronavirus, such as CM and Cataluña), with there being a few responses from other provinces in which the saturation of the ICUs was not so significant.

## 5. Conclusions

Due to the severity and the characteristics of the COVID-19 pandemic, the ICUs have been hard hit and the number of beds had to be considerably increased to treat a large number of patients requiring invasive ventilatory support and specialized care. The results that were obtained in the sample of Spanish professionals who participated in our study show that there was a significant increase in their workload during this pandemic. In addition, there is evidence of an increase in the proportion of staff with little experience and training who provided adequate care for highly critical patients with very advanced care requirements, and more research is needed in this respect to highlight the need for a specific specialty that includes the necessary competencies for the care of critically ill patients. Secondly, even though a large proportion of participants have experienced symptoms, a small percentage of them have tested positive for coronavirus. This may be because the positivity of all of the patients that were admitted to the unit is known a priori and to the availability of PPE in this type of unit, thus we observed in turn a great disparity between the Spanish regions that were studied. In any case, this research highlights the occupational risk factors that have affected the health professionals in Spanish ICUs and the need to act against them from both an interventional and preventive perspective.

**Supplementary Materials:** The following supporting information can be downloaded at: https://www.mdpi.com/article/10.3390/psych4040064/s1, Table S1: STROBE Statement—Checklist of items that should be included in reports of cross-sectional studies.

**Author Contributions:** Conceptualization, methodology, writing—original draft, F.G.-A.; data curation, visualization, writing—original draft, F.J.G.-H.; formal analysis, data curation, C.P.-P.; formal analysis, writing—review & editing, P.C.-M.; and conceptualization, supervision, F.J.C.-M. All authors have read and agreed to the published version of the manuscript.

**Funding:** No external funding was received for this research.

**Institutional Review Board Statement:** The study was approved by the Hospital Universitario Fundación Alcorcón Clinical Research and Ethics Committee (20/88).

**Informed Consent Statement:** Informed consent was obtained from all subjects involved in the study.

**Data Availability Statement:** The data presented in this study are available on request from the corresponding author. The data are not publicly available due to ethical questions.

**Conflicts of Interest:** The authors declare no conflict of interest.

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
