# Peer review of "Socio-Occupational and Health Conditions in Intensive Care Unit (ICU) Professionals during the COVID-19 Pandemic in Spain"

_psych, doi:10.3390/psych4040064_

Round 1

Reviewer 1 Report

Dear Editor,

Consider the article well-written, objective and enlightening.

Consider the article well-written, objective, and originating. I leave these questions to the authors:

A total of 86 (18.66%) professionals report having  had contact outside work with patients affected by COVID

Who are these patients? where are they from?

In the justification for conducting the study, the term emotional perception is brought. At the end of the study, is it not clear the perception framework adopted? Likewise subjective perception?

I consider that the verb to perceive is not well applied to a quantitative study.

Once they assessed workload, physical symptoms, and diagnostic tests related to COVID-19 disease. I suggest adopting: Assess the relationship between workload, physical symptoms and diagnostic tests or describe I suggest restructuring and decreasing the objective.

Did they follow the guidelines of the Strengthening the Reporting of Observational Studies in Epidemiology (STROBE) guidelines?

Author Response

Manuscript ID: psych-1937913

Title: ICU professionals' perception of personal and work situation during the COVID-19 pandemic

Authors: Fernanda Gil-Almagro, Fernando Jose García-Hedrera, Francisco Javier Carmona-Monge, Cecilia Peñacoba-Puente, Patricia Catalá-Mesón.

Ms. Saowalak Rungruang

Psych

Assistant Editor

E-Mail: rungruang@mdpi.com

Dear Ms. Saowalak Rungruang,

We would like to thank you for your interest in our manuscript entitled "ICU professionals' perception of personal and work situation during the COVID-19 pandemic” and now titled “Socio-occupational and health conditions in Intensive Care Unit (ICU) professionals during the COVID-19 pandemic in Spain” (Manuscript Number: psych-1937913). We appreciate the time that you and the other reviewers have dedicated to reading the manuscript and providing suggestions. Your suggestions have enriched the manuscript considerably. Likewise, we have incorporated all the comments suggested. Following your directions, we have proceeded to revise our manuscript, highlighting the changes by using the track changes mode in MS Word.

At the end of this letter, you will find an explanation of the changes made to the manuscript in accordance with your comments.

Once again, we wish to express our appreciation for the clear improvement of the manuscript made possible by the reviewer and editor’s contributions. We hope the new changes meet their expectations, and we hope that they consider the work apt for publication in Psych.

Please do not hesitate to suggest any further changes. We are at your disposal for anything else you may require.

Best regards,

Reviewer 1

Consider the article well-written, objective, originating and enlightening.

Response: We appreciate the reviewer’s kind words and their positive appraisal of our study.

I leave these questions to the authors:

A total of 86 (18.66%) professionals report having had contact outside work with patients affected by COVID. Who are these patients? where are they from?

Response: There has been a confusion with the terminology. We talk about COVID patients but they are family, friends, neighbours or roommates of the participants. In order to clarify this statement, it has been modified to: “a total of 86 (18.7%) professionals declare having had contact outside of work with people affected by COVID”.

In the justification for conducting the study, the term emotional perception is brought. At the end of the study, is it not clear the perception framework adopted? Likewise subjective perception?I consider that the verb to perceive is not well applied to a quantitative study.

Response: Thanks for your suggestion. We totally agree. Actually, our research focuses on the study of socio-occupational and health variables in a sample of ICU professionals during the pandemic in Spain. In the sentence to which the reviewer refers, we have proceeded to change “emotional perception” for “socio-occupational and health conditions”. Additionally, the entire manuscript has been revised to avoid confusion in this regard, especially as regards the term perception.

Once they assessed workload, physical symptoms, and diagnostic tests related to COVID-19 disease. I suggest adopting: Assess the relationship between workload, physical symptoms and diagnostic tests or describe I suggest restructuring and decreasing the objective.

 Response: Thank you for the recommendation. We totally agree. The objective has been completely modified. The new wording of the objective is as follows:

“The main objective of this study is to analyze the socio-occupational and health conditions of health professionals who have worked in the ICU during the first wave of the pandemic. Specifically, sociodemographic variables such as age, sex, marital status, aspects related to the place of residence or the Spanish autonomous community, among other, will be analyzed. Regarding occupational variables, the years of experience in the ICU, the professional category, the availability of PPE, the workload or the patient/professional ratio are of special interest. In relation to health issues, PCR and serology will be analyzed as well as the symptoms associated with COVID-19.

In addition, with regard to working conditions (availability of PPE, workload, patient/professional ratio), this research aims to analyze the possible differences depending on the Spanish region sampled and the professional category, as well as their relationship with the characteristic symptoms of COVID-19 (myalgias and respiratory distress)”.

Likewise, to facilitate the reading and understanding of the manuscript, the Results section has been subdivided into headings according to the proposed objectives.

Did they follow the guidelines of the Strengthening the Reporting of Observational Studies in Epidemiology (STROBE) guidelines?

Response: Thanks for your suggestion. It has been incorporated into the text that the STROBE guideline (for cross-sectional studies) has been followed for the preparation of the manuscript. The STROBE Statement (Checklist of items that should be included in reports of cross-sectional studies) can be found in the Appendix.

Finally, it should be noted that, thanks to the reflections generated by the reviewers' comments, other changes of interest have been made to the manuscript, including the modification of the title. We hope that these changes will contribute, in the opinion of the reviewers, to a better understanding of the manuscript and to improve its quality.

Reviewer 2 Report

This work describes the personal and work situation of ICU professionals during Covid-19 pandemic period. Using an online questionnaire form, authors have evidenced that these workers have fear about infection for themselves and their families. I think that this work is very well written, just some little English errors to change.

Author Response

Manuscript ID: psych-1937913

Title: ICU professionals' perception of personal and work situation during the COVID-19 pandemic

Authors: Fernanda Gil-Almagro, Fernando Jose García-Hedrera, Francisco Javier Carmona-Monge, Cecilia Peñacoba-Puente, Patricia Catalá-Mesón.

Ms. Saowalak Rungruang

Psych

Assistant Editor

E-Mail: rungruang@mdpi.com

Dear Ms. Saowalak Rungruang,

We would like to thank you for your interest in our manuscript entitled "ICU professionals' perception of personal and work situation during the COVID-19 pandemic” and now titled “Socio-occupational and health conditions in Intensive Care Unit (ICU) professionals during the COVID-19 pandemic in Spain” (Manuscript Number: psych-1937913). We appreciate the time that you and the other reviewers have dedicated to reading the manuscript and providing suggestions. Your suggestions have enriched the manuscript considerably. Likewise, we have incorporated all the comments suggested. Following your directions, we have proceeded to revise our manuscript, highlighting the changes by using the track changes mode in MS Word.

At the end of this letter, you will find an explanation of the changes made to the manuscript in accordance with your comments.

Once again, we wish to express our appreciation for the clear improvement of the manuscript made possible by the reviewer and editor’s contributions. We hope the new changes meet their expectations, and we hope that they consider the work apt for publication in Psych.

Please do not hesitate to suggest any further changes. We are at your disposal for anything else you may require.

Best regards,

Reviewer 2

This work describes the personal and work situation of ICU professionals during Covid-19 pandemic period. Using an online questionnaire form, authors have evidenced that these workers have fear about infection for themselves and their families. I think that this work is very well written, just some little English errors to change.

Response: Thank you very much for your words. The manuscript has been revised by a professional translator.

Finally, it should be noted that, thanks to the reflections generated by the reviewers' comments, other changes of interest have been made to the manuscript, including the modification of the title. We hope that these changes will contribute, in the opinion of the reviewers, to a better understanding of the manuscript and to improve its quality.

Reviewer 3 Report

Line 11 - Write the name (e.g. intensive care units) and appropriate acronym (e.g. ICU) in full the first time (including the abstract), then use only acronyms in the text. Similarly with the abbreviations CCU, PPE, etc.

Line 13 – The method should include where the research was conducted, as Catalonia is mentioned in the results (line 20).

Results - Numbers expressed as percentages should be rounded to whole numbers or to one decimal place.

In some parts, the results of the abstract do not follow the conclusions.

Line 26 - Keywords are not correlated with the abstract. Severe acute respiratory syndrome coronavirus 2 is mentioned for the first time in keywords.

Line 53 – “…increases with age, reaching 26.7% and 10.4% in those over 79 years of age, respectively”. Two results are shown in this sentence, with only one age group.

There are some grammatical errors that require attention (i.e. line 167-168).

Line 180-183. The response rate is not adequately explained in the sentence “Of the 1476 responses to the surveys collected between 16/05/2020 and 30/06/2020, in 10 of them (0.68%) the participants did not give their consent to be included in the study and 490 (33.72%) were answered by personnel who have provided their services in the ICU/AED. Finally, the study sample was comprised of 461 surveys (31.73%).”

Line 187 - The results should not be presented in this way in the text “(M = 39.68 years; SD = 10.57)”.

Line 188 - It is not necessary to write in the text (n = 391 (84.82%)). Only percentages (85% or 84.8%) can be specified, other data are available in the table.

Line 194 - Table 1 - It should include other age groups, not only under 40 years, and both sexes.

The meaning of the abbreviations in the table should be entered below the table (e.g. NCT, PPE).

Line 207-211. When the data are written in the text and not shown in the Table or Figure, they should contain the addition at the end of the sentence or paragraph "(data not shown)".

Line 298 - Results reported in the discussion must first be reported in the research results (e.g. …we found that only 17.93% had generated immunity to the coronavirus).

Author Response

Manuscript ID: psych-1937913

Title: ICU professionals' perception of personal and work situation during the COVID-19 pandemic

Authors: Fernanda Gil-Almagro, Fernando Jose García-Hedrera, Francisco Javier Carmona-Monge, Cecilia Peñacoba-Puente, Patricia Catalá-Mesón.

Ms. Saowalak Rungruang

Psych

Assistant Editor

E-Mail: rungruang@mdpi.com

Dear Ms. Saowalak Rungruang,

We would like to thank you for your interest in our manuscript entitled "ICU professionals' perception of personal and work situation during the COVID-19 pandemic” and now titled “Socio-occupational and health conditions in Intensive Care Unit (ICU) professionals during the COVID-19 pandemic in Spain” (Manuscript Number: psych-1937913). We appreciate the time that you and the other reviewers have dedicated to reading the manuscript and providing suggestions. Your suggestions have enriched the manuscript considerably. Likewise, we have incorporated all the comments suggested. Following your directions, we have proceeded to revise our manuscript, highlighting the changes by using the track changes mode in MS Word.

At the end of this letter, you will find an explanation of the changes made to the manuscript in accordance with your comments.

Once again, we wish to express our appreciation for the clear improvement of the manuscript made possible by the reviewer and editor’s contributions. We hope the new changes meet their expectations, and we hope that they consider the work apt for publication in Psych.

Please do not hesitate to suggest any further changes. We are at your disposal for anything else you may require.

Best regards,

Reviewer 3

Line 11 - Write the name (e.g. intensive care units) and appropriate acronym (e.g. ICU) in full the first time (including the abstract), then use only acronyms in the text. Similarly with the abbreviations CCU, PPE, etc.

Response: Thanks for your suggestion. We have proceeded to make the change suggested.

Line 13 – The method should include where the research was conducted, as Catalonia is mentioned in the results (line 20).

Response: Thank you for your suggestion. Information regarding where the research was conducted has been included in the design section. Specifically, it has been incorporated that the professionals provided the services in Spanish ICUs, including all autonomous communities (regions).

Results - Numbers expressed as percentages should be rounded to whole numbers or to one decimal place.

Response: Thank you for your suggestion. Percentages have been rounded to whole numbers or to one decimal place.

In some parts, the results of the abstract do not follow the conclusions.

Response: Thank you for the recommendation. We totally agree. The entire abstract has been reformulated.

Line 26 - Keywords are not correlated with the abstract. Severe acute respiratory syndrome coronavirus 2 is mentioned for the first time in keywords.

Response: Thank you for your suggestion. The keywords have been modified.

Line 53 – “…increases with age, reaching 26.7% and 10.4% in those over 79 years of age, respectively”. Two results are shown in this sentence, with only one age group.

Response: Thank you for your comment. Indeed, the sentence as it was drafted gave rise to confusion. The wording has been changed to read as follows:

“In addition, the percentage of hospitalizations and deaths due to COVID-19 increases with age, reaching 26.7% (hospitalizations) and 10.4% (deaths) in those over 79 years of age [6]”.

There are some grammatical errors that require attention (i.e. line 167-168).

Response: Thank you for your suggestion. The entire manuscript has been reviewed.

Line 180-183. The response rate is not adequately explained in the sentence “Of the 1476 responses to the surveys collected between 16/05/2020 and 30/06/2020, in 10 of them (0.68%) the participants did not give their consent to be included in the study and 490 (33.72%) were answered by personnel who have provided their services in the ICU/AED. Finally, the study sample was comprised of 461 surveys (31.73%).”

Response: Thanks for your suggestion. We totally agree. The sentence has been modified to improve its understanding. It has been specified that this research is part of a broader research project aimed at all health professionals in the period of the COVID-19 pandemic and not only ICU professionals. The wording has been changed to read as follows:

“This study is part of a larger project that aimed to analyze the health and working conditions of health personnel in Spain during the COVID19 pandemic (specifically between 16/05/2020 and 30/06/2020). A total of 1466 health care workers participated in this project, previously signing the corresponding informed consent. Of these, for this particular study, those who had performed their services in the ICU were selected. Finally, the study sample was comprised of 461 ICU professionals (31.7%) (Figure 1)”.

Due to the characteristics of the sampling (non-probabilistic snowball and convenience sampling), the response rate cannot be established. A minimum sample size of 300 to 450 participants was established, according to the recommendations of Guadagnol and Velicer. This information has been included in the manuscript (“Method” section).

Line 187 - The results should not be presented in this way in the text “(M = 39.68 years; SD = 10.57)”.

Response: Thank you for your suggestion. The presentation of the data has been modified.

Line 188 - It is not necessary to write in the text (n = 391 (84.82%)). Only percentages (85% or 84.8%) can be specified, other data are available in the table.

Response: Thank you for your suggestion. Your recommendation has been incorporated.

Line 194 - Table 1 - It should include other age groups, not only under 40 years, and both sexes.

Response: Thank you for the recommendation. We totally agree. Table 1 has been modified incorporating the missing data. In the case of sex, information regarding the percentage of men (15.2%) has not been included, as it is complementary to the percentage of women, and thus avoids redundant information.

The meaning of the abbreviations in the table should be entered below the table (e.g. NCT, PPE).

Response: Thank you for the recommendation. The meaning of the abbreviations have been entered below the table.

Line 207-211. When the data are written in the text and not shown in the Table or Figure, they should contain the addition at the end of the sentence or paragraph "(data not shown)".

Response: Thank you for your suggestion. We have proceeded to add "(data not shown)" in those paragraphs that referred to results that did not appear in the Table or Figure.

Line 298 - Results reported in the discussion must first be reported in the research results (e.g. …we found that only 17.93% had generated immunity to the coronavirus).

Response: Thank you for the recommendation. We totally agree. The particular result to which the reviewer refers (“we found that only 17.93% had generated immunity to the coronavirus”) is extracted from the Table 3. PCR and serology tests performed on the professionals participating in the study (Serology (+)).  However, an error has been detected and it should say "15%" instead of "17.93%". The error has been corrected.

In addition, and following the appropriate recommendation of the reviewer, we have proceeded to review and reformulate both the Results and Discussion sections to improve coherence in relation to the findings of this investigation.

Finally, it should be noted that, thanks to the reflections generated by the reviewers' comments, other changes of interest have been made to the manuscript, including the modification of the title. We hope that these changes will contribute, in the opinion of the reviewers, to a better understanding of the manuscript and to improve its quality.
